# Engineering yeast for the production of breviscapine by genomic analysis and synthetic biology approaches

Xiaonan Liu[1,2], Jian Cheng[1], Guanghui Zhang[3], Wentao Ding[1], Lijin Duan[1], Jing Yang[3,4], Ling Kui[2,5], Xiaozhi Cheng[1], Jiangxing Ruan[1], Wei Fan[3], Junwen Chen[3], Guangqiang Long[3], Yan Zhao[3], Jing Cai[6], Wen Wang[5,7], Yanhe Ma[1], Yang Dong[3,4], Shengchao Yang[3,4] & Huifeng Jiang[1]

The flavonoid extract from *Erigeron breviscapus*, breviscapine, has increasingly been used to treat cardio- and cerebrovascular diseases in China for more than 30 years, and plant supply of *E. breviscapus* is becoming insufficient to satisfy the growing market demand. Here we report an alternative strategy for the supply of breviscapine by building a yeast cell factory using synthetic biology. We identify two key enzymes in the biosynthetic pathway (flavonoid-7-*O*-glucuronosyltransferase and flavone-6-hydroxylase) from *E. breviscapus* genome and engineer yeast to produce breviscapine from glucose. After metabolic engineering and optimization of fed-batch fermentation, scutellarin and apigenin-7-*O*-glucuronide, two major active ingredients of breviscapine, reach to 108 and 185 mg l$^{-1}$, respectively. Our study not only introduces an alternative source of these valuable compounds, but also provides an example of integrating genomics and synthetic biology knowledge for metabolic engineering of natural compounds.

[1] Key Laboratory of Systems Microbial Biotechnology, Tianjin Institute of Industrial Biotechnology, Chinese Academy of Sciences, Tianjin 300308, China. [2] University of Chinese Academy of Sciences, Beijing 100049, China. [3] State Key Laboratory of Conservation and Utilization of Bio-resources in Yunnan, The Key Laboratory of Medicinal Plant Biology of Yunnan Province, Yunnan Agricultural University, Kunming, Yunnan 650201, China. [4] National & Local Joint Engineering Research Center on Germplasm Utilization & Innovation of Chinese Medicinal Materials in Southwestern China, Kunming, Yunnan 650201, China. [5] State Key Laboratory of Genetic Resources and Evolution, Kunming Institute of Zoology, Chinese Academy of Sciences, Kunming, Yunnan 650223, China. [6] State Key Laboratory of Quality Research in Chinese Medicine, Institute of Chinese Medical Sciences, University of Macau, Taipa, Macau, China. [7] Center for Ecological and Environmental Sciences, Northwestern Polytechnical University, Xi'an 710072, China. Xiaonan Liu, Jian Cheng and Guanghui Zhang contributed equally to this work. Correspondence and requests for materials should be addressed to S.Y. (email: 13099437499@163.com) or to H.J. (email: jiang_hf@tib.cas.cn)

The traditional Chinese medicinal plant *Erigeron breviscapus* has been used for more than 1,000 years[1]. Breviscapine, the total flavonoid extract of *E. breviscapus*, was classified as a prescription drug of Chinese medicine 30 years ago for the clinical treatment of cardiovascular and cerebrovascular diseases[2], which are the leading causes of death and disability for older people around the world[3]. Significantly, it has been proven to be effective in dilating vessels, inhibiting platelet aggregation, promoting blood circulation, alleviating myocardial ischemia-reperfusion injury, removing blood stasis and reducing oxidative damage[4–7]. At present, more than ten million patients use breviscapine and related drugs each year in China (Supplementary Note 1). Although 1200 ha are under cultivation with *E. breviscapus* annually, yielding about 4,500 tons of stover, agricultural producers are not able to fulfill the ever growing demand of the Chinese market[8]. It is therefore urgent to develop an alternative, sustainable way to secure the breviscapine supply.

In recent years, engineered yeast strains have been used to produce various natural products of plant origin, including artemisinic acid[9,10], ginsenosides[11,12], etoposide aglycone[13], and opioids[14]. Furthermore, biosynthetic pathways of flavonoids such as naringenin[15] and apigenin[16] have also been engineered into yeasts. The biosynthetic pathway from L-phenylalanine to apigenin has been well characterized[17], which comprises six consecutive steps, catalyzed by phenylalanine ammonia-lyase, cinnamate-4-hydroxylase, 4-coumaroyl-CoA-ligase, chalcone synthase, chalcone isomerase, and flavone synthase II. Although considerable efforts have been made to identify the key enzymes in the biosynthetic pathway of breviscapine[18,19], the complete pathway from apigenin to breviscapine in *E. breviscapus* was still unclear before this study.

In this work, we set out to decode the biosynthetic pathway of breviscapine by combining genomic analysis and synthetic biology tools. Breviscapine mainly contains scutellarin, along with a small amount of apigenin 7-O-glucuronide[18]. Using genomic analysis, we identified two key enzymes flavonoid-7-O-glucuronosyltransferase (F7GAT), which converts apigenin into apigenin-7-O-glucuronide, and flavone-6-hydroxylase (F6H), which functions together with F7GAT to produce scutellarin from apigenin. Subsequently, we constructed a yeast cell factory to produce these compounds of breviscapine from glucose using synthetic biology tools. This work provides insights for the identification of biosynthetic pathways of natural products from traditional Chinese medicinal herbs and illustrates the potential of yeast fermentation system in producing natural compounds through synthetic biology approach.

## Results

**Construction of apigenin-producing platform.** In order to decipher the biosynthetic pathway of breviscapine, the draft genome of *E. breviscapus* was re-annotated by combination of genomic and transcriptomic data (Methods). Multiple copies of the known genes in the flavonoid biosynthesis pathways from phenylalanine to apigenin in *E. breviscapus* were identified based on sequence homology[20–22] (Supplementary Table 1). The candidate genes for hypothetical pathways from apigenin to the two major components of breviscapine were proposed (Fig. 1a). One component of breviscapine is apigenin-7-O-glucuronide, which was expected to be synthesized from apigenin by F7GAT. Another major component of breviscapine is scutellarin, which can plausibly be produced by two putative pathways. One would comprise hydroxylation by F6H to yield the aglycone scutellarein, followed by glycosylation catalyzed by F7GAT, while the other would comprise glycosylation yielding apigenin-7-O-glucuronide, followed by hydroxylation of the flavonoid at position six by F6H.

To confirm our hypothetical pathway to breviscapine, we first proposed to build an engineered yeast platform to characterize functional genes in the genome of *E. breviscapus*. The highly expressed copies for each known gene were chosen to build biosynthetic pathway from phenylalanine to apigenin in yeast (Supplementary Table 2). The engineered DNA fragments with all known genes were integrated into the genome of *Saccharomyces cerevisiae* W303-1B using a modularized two-step chromosome integration approach (Fig. 1b, and Supplementary Tables 3, 4 and Methods)[23]. The product of apigenin in the engineered yeast was confirmed by high-performance liquid chromatography (HPLC) and mass spectrometry (MS) (Supplementary Fig. 1). Thus, we constructed an engineered yeast platform (SC1) to produce precursor apigenin.

**Functional characterization of F7GAT.** F7GATs, which typically transfer UDP-glucuronic acid to the C7 position of flavonoids, have been reported in Lamiales[24–27], but the functional gene had not yet been confirmed in *E. breviscapus*. A total of 83 UDP-glycosyltransferase (UDPGT) genes were identified in the *E. breviscapus* genome and assigned into 15 gene families (Supplementary Fig. 2). Previous studies suggested that enzymes of the UGT88 family can transfer UDP-glucuronic acid to the C7 position of flavonoids[24,26]. Among the identified UDPGT genes, only one belongs to the UGT88 gene family[25,27] (Supplementary Fig. 3), indicating that it may be the F7GAT of *E. breviscapus*. With UDP-glucuronic acid as sugar donor, *EbF7GAT* was indeed able to utilize both apigenin and scutellarein as substrates in vitro (Supplementary Fig. 3). The kinetic parameters of purified *EbF7GAT* were determined for apigenin ($K_m$ 9.24 μM and $k_{cat}$ 0.57 s$^{-1}$) and scutellarein ($K_m$ 70.15 μM and $k_{cat}$ 0.24 s$^{-1}$) (Supplementary Table 5). Furthermore, *EbF7GAT* also can convert many other flavonoid substrates such as naringenin, kaempferol, quercetin, chrysin, baicalein, luteolin, diosmetin, and chrysoeriol at high conversion rates in vitro (Supplementary Fig. 4 and Supplementary Table 6).

In order to confirm the function of EbF7GAT in vivo, we first identified the UDP-glucose dehydrogenase of *E. breviscapus* (EbUDPGDH) based on the orthologous gene from *Glycine max*[28] (Supplementary Fig. 5), which provides substrate in vivo for EbF7GAT. Then a vector carrying *EbF7GAT* and *EbUDPGDH* was introduced into the engineered yeast strain SC1. A new chromatographic peak was detected among the fermentation products around 6.5 min, which was consistent with the retention time of the authentic apigenin-7-O-glucuronide reference standard (Fig. 1c). Furthermore, we confirmed that the molecular weight of the compound in the new peak was the same as that of apigenin-7-O-glucuronide by liquid chromatography-MS (LC-MS) (Fig. 1d). After 5 days of fermentation, the engineered yeast produced about 20 mg l$^{-1}$ of apigenin-7-O-glucuronide, of which 26.6% was located within the cells and 73.4% had secreted into the culture broth (Supplementary Table 7).

**Evolution and expression of the P450 genes.** It has been published that *F6H* should belong to the cytochrome P450 family[29–31], which is the largest gene family in plants[32,33]. In order to identify the gene encoding *F6H* in *E. breviscapus*, a total of 312 putative P450 genes were annotated in the *E. breviscapus* genome based on known P450 genes (Supplementary Data 1 and Methods). To narrow down the list of candidates, we used a strategy of combining chemotaxonomy with evolutionary genomics. We analyzed the distribution of scutellarin production in five sequenced species in the Asteraceae family and eight sequenced species from three other representative plant taxa (Fig. 2a). All of the five studied species from the Asteraceae family (*E. breviscapus*, *Conyza canadensis*, *Lactuca sativa*, *Cynara cardunculus*, and

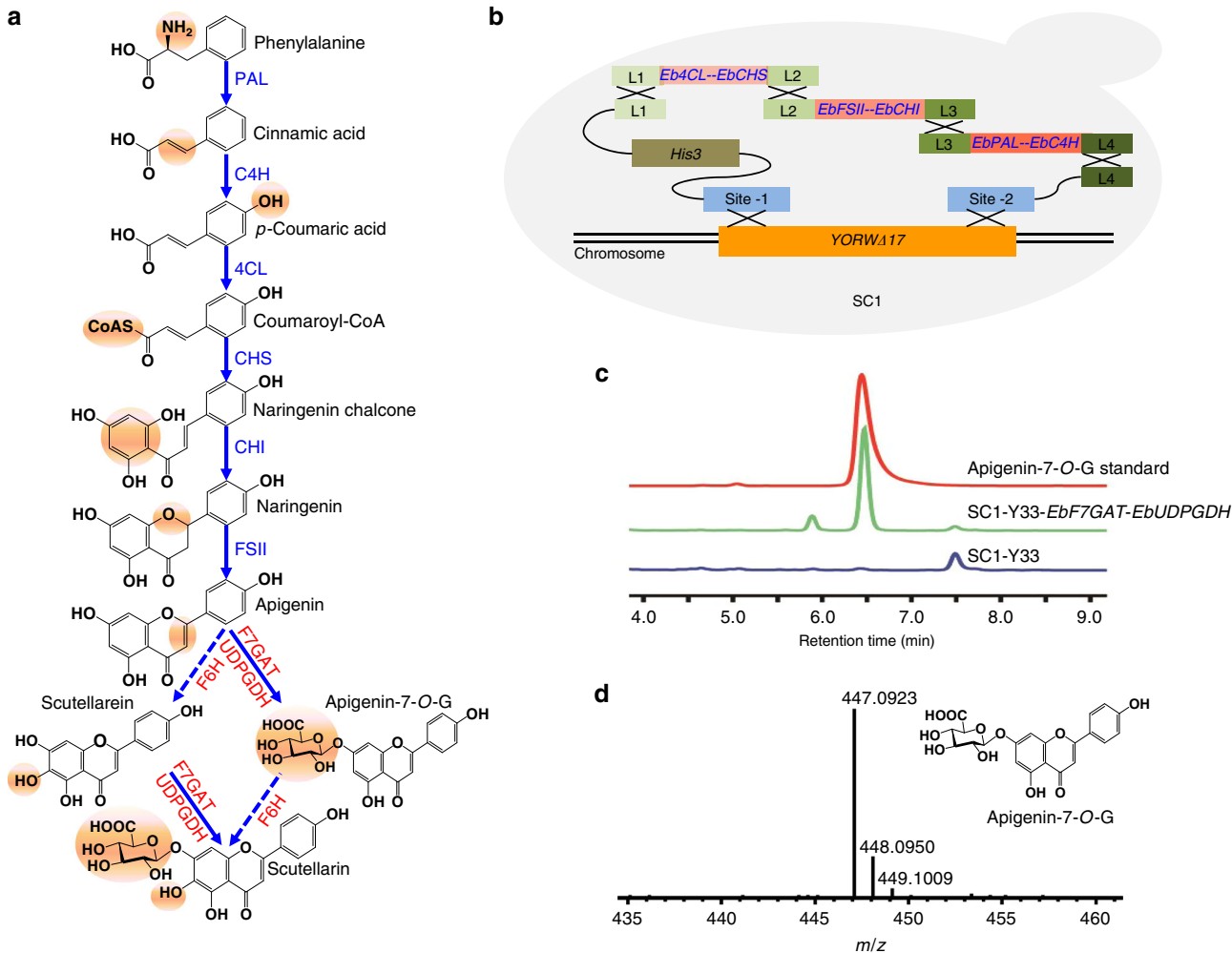

**Fig. 1** Proposed pathway for breviscapine biosynthesis in *E. breviscapus* and construction of the apigenin-7-*O*-glucuronide producing platform strain. **a** Proposed pathway for breviscapine biosynthesis in *E. breviscapus*. Genes that have been reported are shown in blue and those which were identified in this study are shown in red. Abbreviations: apigenin-7-*O*-G, apigenin-7-*O*-glucuronide; 4CL, 4-coumaroyl-CoA ligase; C4H, cinnamate 4-hydroxylase; CHI, chalcone isomerase; CHS, chalcone synthase; FSII, flavone synthase II; F6H, flavone-6-hydroxylase; F7GAT, flavonoid-7-*O*-glucuronosyltransferase; PAL, phenylalanine ammonia lyase; UDPGDH, UDP-glucose dehydrogenase. **b** Construction of the apigenin-producing strain SC1. **c** HPLC analysis of the apigenin-7-*O*-glucuronide standard, the fermented products of the apigenin-7-*O*-glucuronide producing strain (SC1-Y33-*EbF7GAT-EbUDPGDH*) and the negative control strain (SC1-Y33). Y33 is the abbreviation of plasmid YCPlac33. **d** LC-MS analysis of apigenin-7-*O*-glucuronide in the fermented products of SC1-Y33-*EbF7GAT-EbUDPGDH*

*Carthamus tinctorius*) produced scutellarin, but we did not detect scutellarin in non-Asteraceae species (Fig. 2b). These pieces of evidence indicated that the encoding gene of F6H in *E. breviscapus* may be an Asteraceae-specific gene.

In order to identify Asteraceae-specific P450 genes, we annotated all P450 genes in the five sequenced species from the Asteraceae family. Using eight non-Asteraceae species as the out group, we identified 134 Asteraceae-specific P450 genes in the *E. breviscapus* genome (Fig. 2c). However, after comparing the expression levels of all genes in the wild-type species and breviscapine-overproducing cultivars of *E. breviscapus*[34], we did not find significant differences between the Asteraceae-specific and non-Asteraceae-specific P450 genes (Fig. 2d). Interestingly, we found that genes in the flavonoid biosynthetic pathway showed very high gene expression levels in both the wild type and the cultivated breeds. It is therefore possible that *F6H* may also be highly expressed in *E. breviscapus*.

**Functional identification of F6H.** To identify the gene encoding F6H from dozens of candidates, we developed a functional screening method for P450 genes (Fig. 3a and Methods). Recombinant vectors harboring each of 36 highly expressed candidate genes and a cytochrome P450 reductase gene (*EbCPR*), which was cloned from *E. breviscapus* to transport electrons for P450 catalysis, were constructed by Gibson assembly[35]. The resulting vectors were transferred into the engineered yeast strain (SC1-FU) that produces apigenin-7-*O*-glucuronide (Supplementary Table 4). After fermentation for 5 days in baffled 24-well plates, metabolic products of the engineered yeasts were analyzed by HPLC. We screened the putative F6H candidate genes with priorities according to their expression levels. Fortunately, a highly expressed gene, *CYP706X*, induced a new peak with a retention time of 4.6 min, which was the same as that of the authentic scutellarin standard (Fig. 3b). Moreover, the molecular weight of the compound in the new peak was also the same as that of scutellarin, as determined by LC-MS (Fig. 3c). Furthermore, when the vector containing *CYP706X* and *EbCPR* was transferred into our starting strain W303-1B, the engineered yeast was able to convert apigenin into its 6-hydroxy derivative scutellarein (Supplementary Fig. 6a), whose structure was confirmed

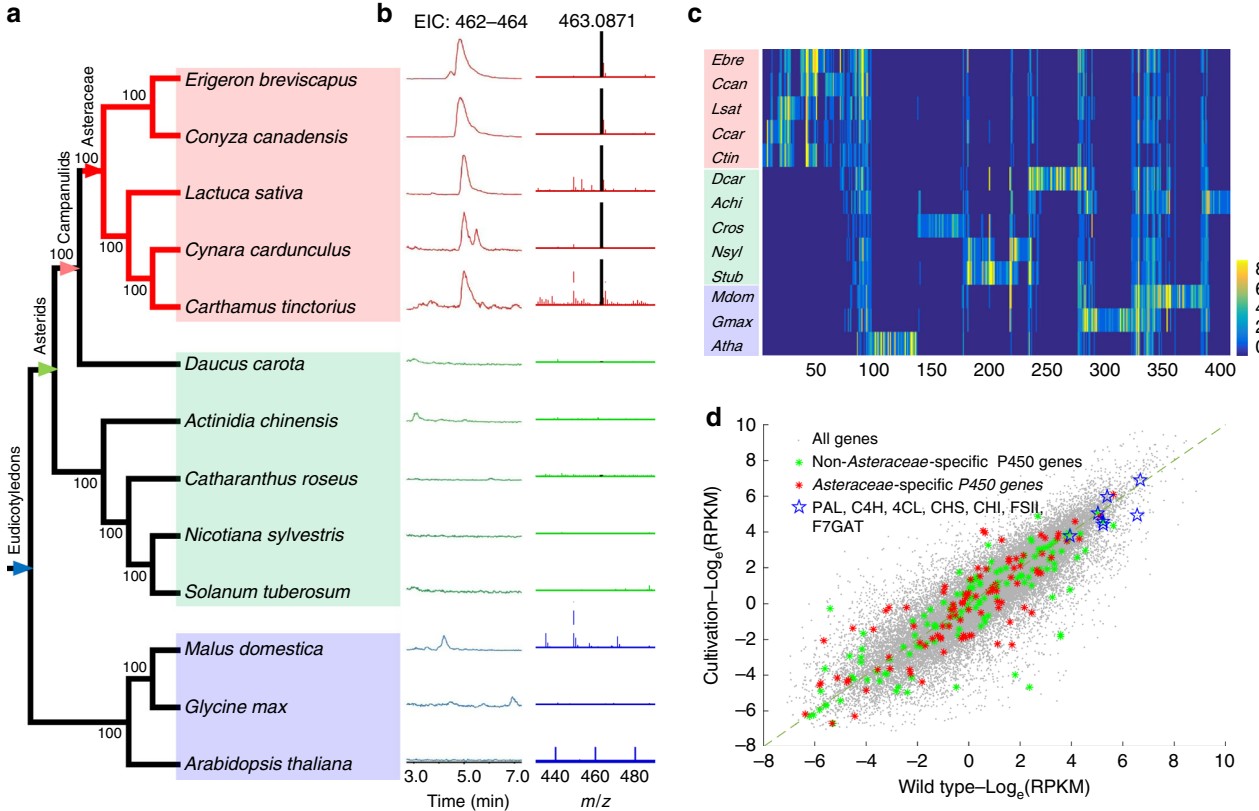

**Fig. 2** Analysis of scutellarin extracted from 13 species and the classification and expression of P450 genes. **a** The phylogenetic relationship among 13 species. The neighbor-joining tree was constructed based on 369 one-to-one orthologous genes in all studied species. All nodes received bootstrap support values from 100 replicates. **b** LC-MS analysis of flavonoids extracted from thirteen plants at 335 nm with molecular masses from 462 to 464 by extracted ion chromatographic (EIC) analysis. The molecular mass of scutellarin was highlighted. **c** The distribution of P450 genes from 13 plants in 411 P450 subfamilies. Each row indicates the plant species and each column indicates the P450 subfamily. The color scale from 0 to 8 indicates the number of P450 genes in a subfamily. **d** The correlation of gene expression in both the cultivated types and wild-type species. The average of gene expression in six cultivated types (Y axis) and in five wild types (X axis) is shown. Gray dots: all genes in E. breviscapus, green asterisks: the non-Asteraceae specific P450 genes, red asterisks: the Asteraceae-specific p450 genes, blue pentagrams: key enzymes in the breviscapine biosynthesis pathway

by LC-MS (Supplementary Fig. 6b) and nuclear magnetic resonance (NMR) (Methods and Supplementary Fig. 7). However, the engineered yeast was not able to convert apigenin-7-O-glucuronide into its 6-hydroxy derivative scutellarin in vivo (Supplementary Fig. 6c).

In order to further confirm the function of *CYP706X* in vitro, we proposed to isolate microsomal proteins from the engineered strains expressing *CYP706X* and *EbCPR*. In vitro assays of microsomal proteins were conducted using apigenin or apigenin-7-O-glucuronide as the substrate. We found that the microsomal protein fractions containing *CYP706X* and *EbCPR* were able to catalyze the hydroxylation of apigenin at position six, but could not convert the hydroxyl group to apigenin-7-O-glucuronide (Fig. 3d, e). Thus, the P450 gene *CYP706X* was identified as *EbF6H*. Furthermore, we speculated that apigenin should first be converted into scutellarein by *EbF6H*, after which scutellarein could be converted into scutellarin by *EbF7GAT* in *E. breviscapus*.

**Biosynthesis of breviscapine in yeast**. Although we were able to produce both components of breviscapine when we introduced all genes of the corresponding biosynthetic pathways into yeast, the yield was still very low, with only 9.2 mg of scutellarin per gram of dry cell weight (DCW). To overcome this problem, we further engineered the host cells' metabolism to increase the breviscapine yield. As three molecules of malonyl-CoA are needed to form the flavonoid backbone[17], we surmised it would be helpful to improve the intracellular concentration of malonyl-CoA (Fig. 4a). As the

concentration of acetyl-CoA, the immediate precursor of malonyl-CoA, undergoes dynamic changes due to continuous generation and consumption, we first tried to decrease the consumption of acetyl-CoA by deleting the cytosolic malate synthase gene (*MLS1*), and thus preventing the associated oxidation of acetyl-CoA[36]. This improved the yield of scutellarin from 9.2 to 11.6 mg g$^{-1}$ DCW. In the next optimization step, we deleted the peroxisomal citrate synthase gene (*CIT2*) in the yeast strain, which has been published to block the consumption of acetyl-CoA[36]. The scutellarin yield in this double-knockout strain was improved to 14.3 mg g$^{-1}$ DCW (Fig. 4b).

Second, we proposed to increase the supply of acetyl-CoA in the metabolic network. Overexpression of the endogenous alcohol dehydrogenase gene (*ADH2*) can lead to the conversion of more ethanol into acetaldehyde and acetyl-CoA[37]. Moreover, overexpression of the endogenous aldehyde dehydrogenase gene (*ALD6*) and introduction of an acetyl-CoA synthetase variant from *S. enterica* ($ACS_{SE}^{L641P}$) was reported to increase the metabolic flux towards malonyl-CoA[38]. We designed a malonyl-CoA-producing multiple-gene module that integrated the genes *ALD6*, $ACS_{SE}^{L641P}$, and *ADH2* into the *YPRCΔ15* locus of the double-knockout strain. The resulting engineered stain was able to produce 15.5 mg g$^{-1}$ DCW of scutellarin from 2% glucose in flask cultures. Thus, we obtained an engineered yeast strain with a 1.68-fold improved scutellarin yield compared with the starting strain (Fig. 4b).

Finally, the engineered strain was tested in scaled-up fermentations in a 3-L benchtop fermenter. During the

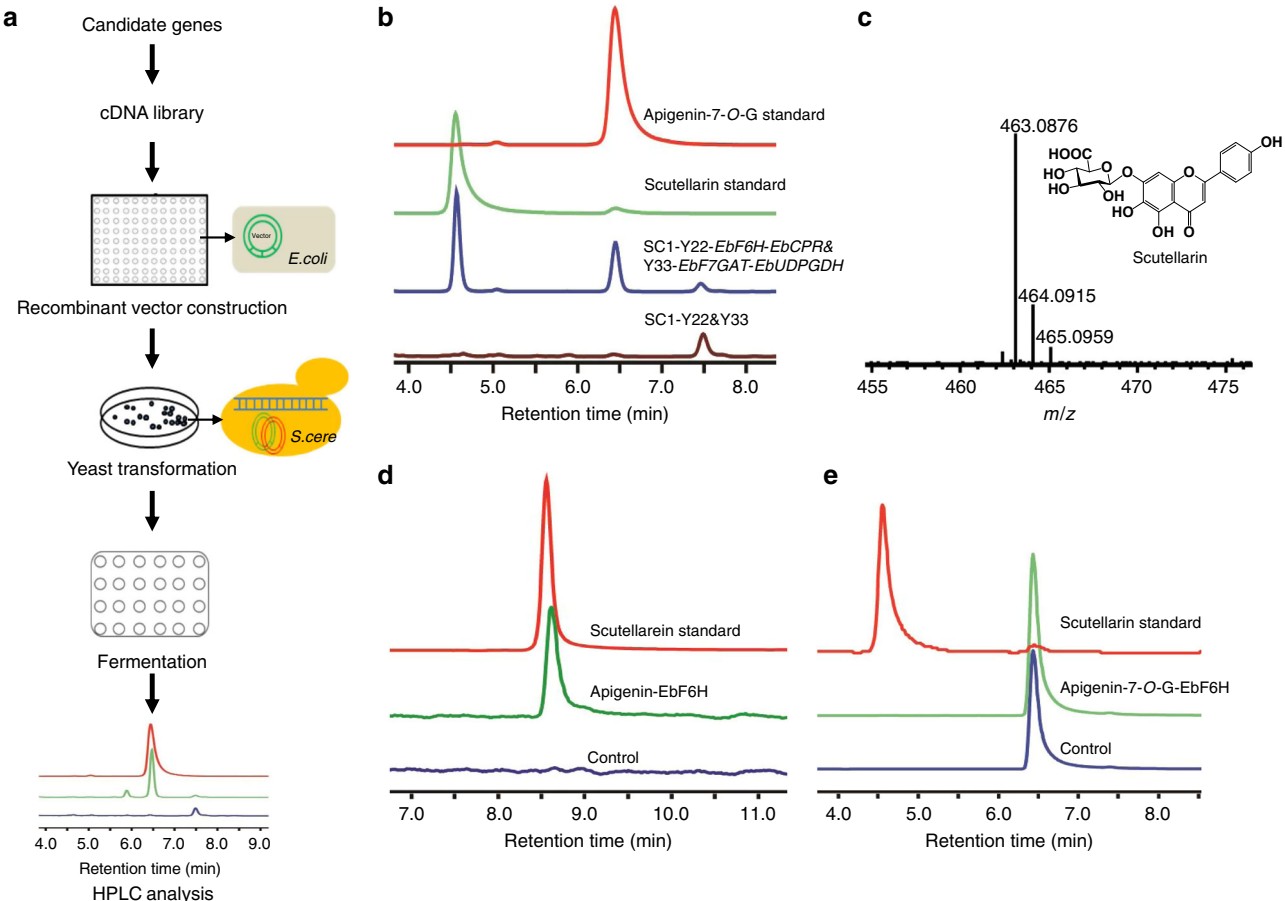

**Fig. 3** P450 enzymes screening and functional identification. **a** Overview of the P450 gene screening method. The detailed method is described in Methods. **b** HPLC analysis of the apigenin-7-O-glucuronide standard, scutellarin standard, the fermented product of the scutellarin-producing strain (SC1-Y22-*EbF6H-EbCPR*&Y33-*EbF7GAT-EbUDPGDH*), and the negative control strain (SC1-Y22&Y33). Y22 and Y33 represent the plasmids Ycplac22 and YcplacY33. **c** MS analysis of scutellarin in the fermented product of the scutellarin-producing strain. **d** HPLC analysis of microsomal enzyme assays (containing *EbF6H* and *EbCPR*) incubated with apigenin in vitro. The microsomal fraction containing only *EbCPR* was used as negative control. **e** HPLC analysis of microsomal enzyme assays incubated with apigenin-7-O-glucuronide in vitro. The microsomal fraction containing only *EbCPR* was used as the negative control

fermentation process, the dissolved oxygen concentration (DOC) is crucial for the proportion of breviscapine in total flavonoids, as *EbF6H* is a P450 enzyme, and therefore requires molecular oxygen for its catalytic reaction[39]. When we optimized the DOC during the fermentation, the maximum yields of scutellarin and apigenin-7-O-glucuronide after 7 days of fed-batch fermentation reached 108 and 185 mg l$^{-1}$, respectively (Fig. 4c). Although the obtained yield is already quite high compared with the starting strain, it should be possible to further increase it via metabolic engineering and fermentation optimization in the future to utilize the full potential of this yeast fermentation system.

## Discussion

P450 enzymes catalyze key steps in the biosynthetic pathways of many highly valuable plant natural products. In our study, we identified 36 new P450 subfamilies in the genome of *E. breviscapus* (Supplementary Data 1). One of subfamilies that performs the F6H function appears to have originated from the ancient P450 gene family *CYP706A*, whose members can catalyze the hydroxylation of terpenoids[40,41]. This subfamily is profoundly different from the reported F6H in other plants, such as the *CYP82D* subfamily from the Lamiaceae *Ocimum basilicum*[31], *Mentha piperita*[31], *Scutellaria baicalensis*[42], and *Salvia*

*miltiorrhiza*[43], as well as the less closely related to *CYP71D9* from *G. max*[30]. The orthologous genes of these reported *F6H* in *E. breviscapus* genome did not perform *F6H* function at all (Supplementary Table 8). Furthermore, based on evolutionary analysis, we found that *CYP706A* was present in four copies in the common ancestor of the Asteraceae (Supplementary Fig. 8). However, only one of the *CYP706X* copies evolved the flavone 6-hydroxylation activity (Supplementary Fig. 8). Interestingly, this function was present in all the studied species of Asteraceae, as the *EbF6H* orthologs from the studied Asteraceae species (*CcF6H*, *LsF6H*, *CcarF6H*, and *CtF6H*) also showed the same function (Supplementary Fig. 9). Moreover, this was consistent with our observations regarding the distribution of the scutellarin production phenotype (Fig. 2b). We therefore speculated that *F6H* may have independently originated in the common ancestor of the Asteraceae, while more research needs to be conducted to illustrate the underlying evolutionary mechanisms. Our study also illustrates the power of combining genomic analysis and synthetic biology knowledge for metabolic engineering of natural compounds. Otherwise, it would be extremely difficult to pinpoint a single novel gene out of several hundred P450 candidates.

The vigorous development of sequencing technologies has enabled great breakthroughs in the identification of biosynthetic pathways of natural products[44]. In this study, we combined

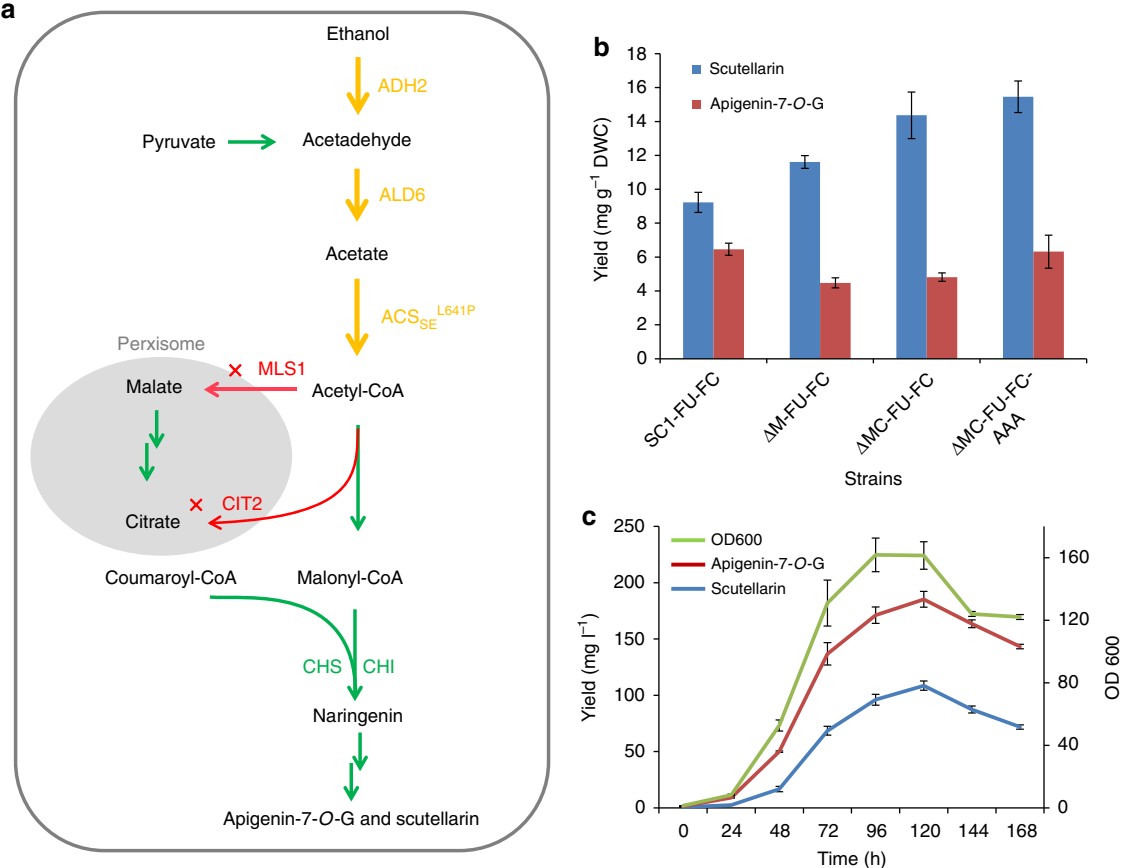

**Fig. 4** Metabolic engineering and fed-batch fermentation. **a** Metabolic engineering to improve the production of malonyl-CoA in the cytosol. Overexpressed genes are shown in orange and deleted genes are shown in red. Abbreviations: $ACS_{SE}^{L641P}$, acetyl-CoA synthase variant from *Salmonella enterica*; ADH2, endogenous alcohol dehydrogenase; ALD6, endogenous aldehyde dehydrogenase from *S. cerevisiae*; CIT2, citrate synthase; MLS1, malate synthase. **b** Evaluation of the engineered yeast strains SC1-FU-FC (SC1-Y22-*EbF6H*-*EbCPR* and Y33-*EbF7GAT*-*EbUDPGDH*), ΔM-FU-FC (SC1-FU-FC-Δmls1), ΔMC-FU-FC (SC1-FU-FC-Δmls1-Δcit2), and ΔM-FU-FC-AAA (ΔMC-FU-FC-ACS-ALD6-ADH2) (more details in Supplementary Table 4) for apigenin-7-*O*-glucuronide and scutellarin production in shake flask fermentations. **c** The production of scutellarin and apigenin-7-*O*-glucuronide by strain ΔM-FU-FC-AAA in fed-batch fermentation. Three replicates were performed for each analysis and the error bars represented the SD

genomic and transcriptomic technologies to uncover the complete biosynthetic pathways of two major components of breviscapine—scutellarin and apigenin-7-*O*-glucuronide. In this pathway, the precursor apigenin is directly converted into apigenin-7-*O*-glucuronide by *EbF7GAT* and can also be used to synthesize scutellarin by *EbF6H* and *EbF7GAT*. Accordingly, both compounds were produced by an engineered yeast strain expressing *EbF6H* and *EbF7GAT*. Currently, the total yield of breviscapine in the engineered yeast is about 300 mg l$^{-1}$ (the sum of 108 mg l$^{-1}$ scutellarin and 185 mg l$^{-1}$ apigenin-7-*O*-glucuronide), which is comparable to the yields of many other flavonoids such as naringenin[15,45] and apigenin[16,46]. Moreover, it is feasible to modulate the ratio of scutellarin to apigenin-7-*O*-glucuronide in the final fermentation product according to the real ratio of both compounds in breviscapine by regulating the gene expression levels of *EbF6H* and *EbF7GAT* in engineered yeasts. It should therefore be possible to directly produce the drug breviscapine using engineered yeasts in future.

To our best knowledge, this work represents the first biosynthesis of breviscapine in a heterologous host starting from glucose. By combining genomic and transcriptomic analysis and high-throughput functional screening, we identified all the enzymes of the breviscapine biosynthetic pathway in *E. breviscapus*, including *EbF6H* and *EbF7GAT*. Our work also illustrates the potential of the yeast fermentation system to increase the yield of breviscapine through metabolic engineering and the

optimization of fermentation conditions. Moreover, this work provides new insights for the identification of biosynthetic pathways of natural products from thousands of traditional Chinese medicinal herbs and a tremendous potential to cheaply produce these compounds through synthetic biology.

## Methods

**Gene re-annotation.** The draft genome of *E. breviscapus* was released as Data Note in our previous study[47]. However, the total genome assembly is ~ 80% of the estimated genome size. In order to detect all genes in the biosynthetic pathway of breviscapine, we re-annotated genes in *E. breviscapus* by combining the genomic data and the transcriptomic data, which include 40.73 GB RNA sequencing (RNA-seq) data from the leaves of five wild individuals, 39.74 GB RNA-seq data from the leaves of six cultivated individuals and 37.03 GB RNA-seq data from the leaf, root, stem, and flower tissues of a cultivated individual[47]. These RNA-seq data were respectively assembled to predict the protein-coding genes. Then all predicted genes were merged together to form a comprehensive and non-redundant reference gene set using OrthoMCL[48] and EVidenceModeler[49]. Finally, 51,715 protein-coding genes in *E. breviscapus* were annotated (Supplementary Data 2), which performed about 96% completeness using BUSCO[50].

**Annotation and analysis of P450 genes.** The P450 genes in the *E. breviscapus* genome were predicted using hmmsearch in conjunction with the P450 hmm model PF00067 ($E$-value < 1E$^{-10}$) from Pfam[51,52]. Putative P450 proteins were screened by amino acid length (350 < length < 650). A total of 312 P450 genes were predicted in the *E. breviscapus* genome (Supplementary Data 1).

In order to analyze the evolutionary process of the P450 gene family in the *E. breviscapus* genome, we proposed to compare the P450 genes of *E. breviscapus* with those of its close relatives and other typical eudicotyledons. Four extra Asteraceae

                                                                                                                   

species (i.e., *C. canadensis*, *L. sativa*, *C. cardunculus*, and *C. tinctorius*), and eight non-Asteraceae species that are typical representatives of seven eudicotyledons orders (i.e., *Catharanthus roseus*, *Daucus carota*, *Actinidia chinensis*, *Nicotiana sylvestris*, *Solanum tuberosum*, *Malus domestica*, *G. max*, and *Arabidopsis thaliana*) were used. The protein sequences of *D. carota*, *N. sylvestris*, *S. tuberosum*, *M. domestica*, *G. max*, and *A. thaliana* were directly downloaded from NCBI website (www.ncbi.nlm.nih.gov). The protein sequences of *C. canadensis*, *L. sativa*, *C. cardunculus*, *C. tinctorius*, *A. chinensis*, and *C. roseus* were predicted using AUGUSTUS[53], based on genome sequences downloaded from NCBI website. All potential P450 proteins in the twelve species were respectively predicted using the same process above. We found a total of 3663 P450 proteins in thirteen plant genomes (including *E. breviscapus*, Supplementary Table 9).

All 3,663 P450 proteins were further classified into 411 P450 subfamilies based on 2 criteria: (1) the proteins in a subfamily had relatively closer phylogenetic relationship in the phylogenetic tree, which was constructed by the alignment of all P450 proteins; (2) the identity between two protein sequences in a subfamily was higher than 55%[54]. Among 411 P450 subfamilies, the subfamilies only including P450 proteins in 5 Asteraceae species, were regarded as Asteraceae-specific P450 subfamilies. All 312 P450 enzymes in the *E. breviscapus* genome could be classified into 60 non-Asteraceae specific sub-families including 178 P450 genes and 36 Asteraceae-specific sub-families including 134 P450 genes (Supplementary Data 1). It should be noted that "Asteraceae-specific" is a relative definition, as there were only eight non-Asteraceae species in the outgroup, so that some bona fide Asteraceae-specific P450 proteins in our study may be present in other non-Asteraceae species. The true Asteraceae-specific P450 proteins are thus likely a subset of our bona fide Asteraceae-specific P450 proteins.

**Extraction of flavonoids**. Plant materials from *E. breviscapus*, *C. canadensis*, *L. sativa*, *C. cardunculus*, *C. tinctorius*, *A. chinensis*, *D. carota*, *C. roseus*, *N. Sylvestris*, *S. tuberosum*, *M. domestica*, *G. max*, and *A. thaliana* were obtained from Yunnan Agricultural University. For preparation of flavonoids, the plant materials were ground in a mortar under liquid nitrogen. The resulting powder was dissolved in 80% methanol (W/V = 1:5), ultrasonicated for 30 min and the cell debris was removed by centrifugation at $13,000 \times g$ for 30 min. The supernatant was used for HPLC analysis.

**Chemicals and media**. Yeast nitrogen base without amino acids and ammonium sulfate (YNB), Bacto peptone, Bacto yeast extract, Luria Broth (LB), agar, lithium acetate, ssDNA, and glucose were obtained from Solarbio, China. Kanamycin, ampicillin, amino acids, adenine, histidine, leucine, tryptophan, and uracil were obtained from Sigma-Aldrich, USA. Chromatography-grade methanol and acetonitrile were obtained from EMD Chemicals, USA. Chromatography-grade formic acid and isopropanol were obtained from Thermo Fisher Scientific, USA. Authentic reference standards of apigenin, apigenin-7-*O*-glucuronide, scutellarin, and scutellarein were obtained from Solarbio. Restriction enzymes, DNA polymerase, and dNTPs were purchased from Thermo Fisher Scientific. T4 ligase, *Bsa*I, and $100 \times$ bovine serum albumin (BSA) were purchased from NEB, USA. *S. cerevisiae* W303-1B (*MATα leu2-3112 ura3-1 trp1-92 his3-11,15 ade2-1 can1-100*)[55] was used as the parent strain for all engineered strains. Competent cells of *E. coli* Trans-T1 (TransGen Biotech, China) were used for recombinant vectors construction and *E. coli* BL21 (*DE3*) (TransGen Biotech) was used for protein expression. *E. coli* was grown in LB medium with appropriate antibiotics. Yeast strains were grown either in SD (Synthetic Dropout) medium lacking leucine, uracil, tryptophan, and histidine, or in Yeast Extract Peptone Dextrose medium[56]. SD medium was used for shake-flask fermentation of yeast. All yeast strains were grown 15 ml culture tubes containing 4 ml medium and at 30 °C 200 r.p.m. to an $OD_{600}$ of ~ 1.0. Flasks (250 ml) containing 50 ml of medium were then inoculated to an $OD_{600}$ 0.05 using the resulting seed cultures. The main cultures were grown at 30 °C and 250 r.p.m. for 5 days and used for HPLC analysis.

**Construction of apigenin producing yeast strain**. To construct an apigenin producing strain, the corresponding biosynthetic pathway was introduced into yeast. The *EbPAL*, *EbC4H*, *Eb4CL*, *EbCHS*, *EbCHI*, and *EbFSII* coding sequences from *E. breviscapus* were synthesized by GENWIZ with codon optimization for *S. cerevisiae*. The M2S integration method was applied for gene cloning and chromosome integration[23]. Briefly, *Eb4CL* and *EbCHS* were amplified with the addition of a *Bsa*I digestion site and ligated with head-to-head promoters (*pADH1-pHXT7*) into the bi-terminator vector T1-(*tTPI1-tPGI*), resulting in the plasmid T1-*Eb4CL*-*EbCHS*. The head-to-head promoters were assembled by overlap PCR with *Bsa*I restriction enzyme sites flanking the ends. The construction of the bi-terminator vectors has been described by Li et al.[23]. Two terminators were inserted into the scaffold plasmid, with dedicated homologous arms (L1 and L2) lying on both sides. The plasmids T2-*EbCHI*-*EbFSII* and T3-*EbPAL*-*EbC4H* were constructed analogously. The integration site YORWΔ17 was chosen as the target locus and *His3* as the selection marker. Each expression cassette with designed homologous arms (primers: L1-F/L2-R, L2-F/L3-R, L3-F/L4-R) was amplified individually. The selection marker module and integration homologous arm module (17Site1-His3-L1 and L4-17site2) were also amplified. All the amplified fragments were used to co-transform W303-1B for assembly and integration, and transformants were selected on a histidine-minus plate (SD-His). Positive transformants were verified

by sequencing, yielding the strain SC1 (Fig.1b). The DNA sequences used in this study are listed in Supplementary Data 3. All strains are listed in Supplementary Table 4. All primers used in vectors construction are listed in Supplementary Table 10.

**Construction of apigenin-7-*O*-glucuronide producing strain**. To identify the *F7GAT* gene in *E. breviscapus*, 83 *UDPGT* genes were first identified by the *UDPGT* hmm model PF00201 (*E*-value < 1E$^{-10}$) from Pfam[51,52], among which 78 *UDPGT* genes could be assigned to 15 gene families. There was only one *UDPGT* gene belonging to the *UGT88* gene family (Supplementary Fig. 3a) and this sequence (Ebre_g070227) was considered as the potential *F7GAT* of *E. breviscapus*. The coding gene of *EbF7GAT* was amplified using the primers EbF7GAT-GG-5F/3 R. The *UDPGDH* gene of *E. breviscapus* (Ebre_g026582) was predicted based on the orthologous gene in *G. max*[28]. The coding gene of *EbUDPGDH* was amplified using the primers EbUDPGDH-GG-5F/3 R.

The coding genes of *EbF7GAT* and *EbUDPGDH* were cloned into the centromeric vector YCPlac33 using Golden Gate assembly. First, the terminator *tGPD1* and *tPFK1* with *Bsa*I digestion sites were fused and integrated into the plasmid YCPlac33, resulting in Y33-GP. Second, head-to-head promoters (*pTDH3-pADH1*) were assembled by overlap PCR with *Bsa*I restriction enzyme sites flanking the ends. Finally, the amplified fragments of *EbF7GAT*, *EbUDPGDH*, and the head-to-head promoters (*pTDH3-pADH1*) were cloned into the vector Y33-GP through Golden Gate cloning[23], resulting in the plasmid Y33-*EbF7GAT*-*EbUDPGDH*. This expression plasmid was introduced into the strain SC1 and generated SC1-FU for product identification.

**P450 gene screening method**. By analyzing the genome of *E. breviscapus*, 36 highly expressed P450 genes were found as *F6H* candidates. First, the terminators *tTDH1* and *tCYC1* with *Bsa*I digestion sites were fused and integrated into the plasmid YCPlac22, resulting in Y22-TC. Second, a candidate gene (Ebre_g067055), *EbCPR*, and the head-to-head promoters (*pPGK1-pTDH3*) were cloned into the vector Y22-TC through Golden Gate cloning. The resulting plasmid harbored the expression cassette of P450 and *EbCPR*. Then, the P450 gene in the vector was substituted by other candidates through Gibson assembly[35]. All the resulting P450 expression plasmids were individually introduced into SC1-FU for product identification. All primers used for P450 gene screening are listed in Supplementary Table 11.

Three colonies were picked for each genotype and used to inoculate 2 ml of SD medium (minus histidine, tryptophan, and uracil) in a 24-well plate. The cells were grown in a shaker at 30 °C and 800 r.p.m. for 48 h, after which the resulting seed cultures were transferred into fresh medium at a ratio of 1:50 and fermented under the same condition for 5 days. The products were analyzed by HPLC.

**Metabolic engineering for scutellarin production**. *MLS1* and *CIT2*, encoding enzymes for acetyl-CoA consumption, were selected for deletion in this study. For gene knockout, the *Ura3* selectable marker was first amplified from YCplac33 and flanked with 1,000 bp upstream and downstream homologous sequences of the target genes. The resulting fragment was integrated into the yeast genome by homologous recombination. Then a fragment with fused 1,000 bp of upstream and downstream homologous sequences was integrated into the genome by homologous recombination to rescue the *Ura3* marker; 5-fluoroortic acid[57] was added to the SD plate to select for the loss of the *Ura3* marker. All primers used for yeast genomic knockouts are listed in Supplementary Table 12.

*ACS*$_{Se}$$^{P641L}$, *ALD6*, and *ADH2*, encoding an acetyl-CoA synthesis pathway from ethanol, were overexpressed to enhance the acetyl-CoA supply for flavonoid production. *ACS*$_{Se}$$^{P641L}$ form *S. enterica* was synthesized by GENEWIZ, and *ALD6* and *ADH2* were amplified from the genome of W303-1B. *ACS*$_{Se}$$^{P641L}$ and *ALD6* were cloned into the vector T4 via Golden Gate assembly, and *ADH2* was analogously cloned into the vector T5. *ACS*$_{Se}$$^{P641L}$, *ALD6*, and *ADH2* were expressed from *pTEF2*, *pPGK1*, and *pHXT7* respectively (primers: L4-F/L5-R, L5-F/L6-R).The expression cassettes of these genes were amplified with flanking sequences and co-transformed with a flanked *Leu2* (15site1-Leu2-L4 and L6-15site2) gene into ΔMC-FU-FC, with genomic integration at the YORWΔ15 site, resulting in the strain ΔMC-FU-FC-AAA.

**EbF7GAT purification and enzyme assays**. The *EbF7GAT* coding gene was ligated into the expression vector pET-28a via NdeI and XhoI restriction sites to construct the plasmid pET28a-*EbF7GAT*, and the plasmid was transformed into *E. coli* BL21 (*DE3*). The engineered strain was grown to an $OD_{600}$ of 0.6 at 37 °C in LB medium supplemented with 100 μg ml$^{-1}$ of kanamycin and then induced at 16 °Cby the addition of IPTG (isopropyl-β-ᴅ-thiogalactopyranoside) to a final concentration of 0.5 mM, and continued overnight. The cells were harvested by centrifugation at $13,000 \times g$ for 30 min and stored at − 80 °C until analysis.

The frozen cell paste was thawed on ice and suspended in 35 ml of lysis buffer (50 mM Tris-HCl, pH 7.5). The cells were disrupted on ice using a high-pressure homogenizer (JNBIO, China), and the cell debris was removed by centrifugation at $13,000 \times g$ for 30 min. To bind the recombinant enzyme, which was expressed as a fusion protein containing the 6-His tag, the supernatant was passed through a 0.45 μm filter and loaded onto a Ni$^{2+}$-chelating affinity chromatography column (GE

Healthcare, USA). After the column was rinsed with 50 ml wash buffer (50 mM Tris-HCl pH 7.5 and 50 mM imidazole), the proteins were eluted with 30 ml elution buffer (50 mM Tris-HCl pH 7.5 and 100 mM imidazole). The eluted proteins were concentrated and dialyzed against lysis buffer (50 mM Tris-HCl pH 7.5) by ultrafiltration using an amicon ultracentrifuge filter device (Millipore, USA). The purity of the protein *EbF7GAT* was evaluated by 12% SDS-PAGE. The protein concentration was determined using a BCA Protein Assay Reagent Kit (Pierce, USA) with BSA as the standard.

The standard reaction mixture (100 μl) consisted of 50 mM Tris-HCl pH 7.5, 100 mM glycosyl acceptor, 1 mM UDPGA, and purified *EbF7GAT* enzyme. After a 10 min pre-incubation of the mixture without the enzyme at 37 °C, the reaction was initiated by the addition of 1 μg of the purified enzyme. After incubation at 37 °C for 8 h, the reaction was stopped by the addition of 100 μl of methanol. The enzyme activity was assessed by measuring the generation of the corresponding glycosylation products via HPLC.

Enzyme kinetics with apigenin as substrate were determined in assays with 0.05 μg of purified recombinant protein, apigenin concentrations of 0.1–20 μM and a reaction time of 10 min; for scutellarein, the substrate was performed in enzyme assays with 0.5 μg of purified recombinant protein and scutellarein concentrations were 5–300 μM. Kinetic parameters were determined from triplicate experiments using GraphPad Prism 5 (GraphPad Software, USA).

**Microsome isolation and enzyme assays**. A single colony of W303-FC or W303-C was used to inoculate a 5 ml tube and incubated at 30 °C for 24 h. A 4 ml aliquot of the resulting culture was transferred to 200 ml of SD medium lacking tryptophan and grown for about 30 h until the $OD_{600}$ reached 2.0. Cells were collected by centrifugation, washed twice with 10 and 5 ml of TEK buffer (50 mM Tris-HCl pH 7.4, 1 mM EDTA, and 0.1 M KCl), disrupted on ice in 30 ml of TES-B buffer (50 mM Tris-HCl pH 7.5, 1 mM EDTA, and 600 mM sorbitol) using a high-pressure homogenizer (JNBIO), and the cell debris was removed by centrifugation at 13,000 × $g$ for 1 h. The supernatant was collected and the pellet was further washed twice with 1 ml of TES-B buffer each time. The combined homogenate was centrifuged for 10 min at 13,000 × $g$ to pellet the mitochondria and nuclei. $CaCl_2$ at a final concentration of 18 mM was added to the supernatant and the resulting mixture incubated for 15 min on ice. Microsomes were harvested by centrifugation for 1 h at 13,000 × $g$, re-suspended in 1 ml of TEG buffer (50 mM Tris-HCl pH 7.4, 1 mM EDTA, and 20% glycerol), and stored at −80 °C. All steps for microsomal preparation were performed at 0–4 °C.

Protein concentrations were determined using a BCA Protein Assay Reagent Kit (Pierce) with BSA as the standard. EbF6H was assayed in 200 ml reaction mixtures containing 100 mM sodium phosphate buffer (pH 7.9), 0.5 mM reduced glutathione, 50 μg of crude protein extract and 200 μM apigenin or apigenin-7-*O*-glucuronide. The assays were initiated by adding 1 mM NADPH and incubated for 30 min, after which 200 μl methanol was added to quench the reactions. Microsomal proteins extracted from yeasts harboring the empty vector were assayed as a negative control.

**Analytical methods**. Optical density was measured at 600 nm using a V-1800 spectrophotometer (Mapada, China). For measurements of flavonoid compounds, 900 μl culture samples were diluted with an equal volume of 100% methanol. After vigorous mixing and ultrasonic breaking for 30 min, the lysates were spun down at 13,000 × $g$ for 10 min. The supernatant was analyzed using a LC-20ADXRHPLC system (Shimadzu, Japan) equipped with a photodiode-array detector. Apigenin, scutellarein, scutellarin, and apigenin-7-*O*-glucuronide were measured at 335 nm using a Kinetex 5 μm Biphenyl 100 Å LC Column (250 × 4.6 mm; Phenomenex, USA) operating at 30 °C. The mobile phase consisted of 0.1% formic acid and acetonitrile with methanol at a flow rate of 1 ml min⁻¹ using the following gradients: 0–20 min, 22% acetonitrile, 5% methanol; 20–22 min, acetonitrile increased from 22 to 90%; 25 min, 22% acetonitrile. Subsequently, the column was washed and equilibrated for 5 min before the next injection. Thirty microliters of the sample were injected into the HPLC system and each run was stopped at 30 min after the injection.

Identification of apigenin, scutellarein, apigenin-7-*O*-glucuronide, and scutellarin was performed using an Agilent 1200 HPLC system coupled with a Bruker-MicrOTOF-II mass spectrometer (Bruker, Germany) equipped with an electrospray ionization (ESI) device. Data acquisition and processing were performed using MicrOTOF control version 3.0/Data Analysis Version 4.0 software. Data were recorded using MassLynx 4.0 software (Waters, USA). All spectra were recorded in positive ion mode over an $m/z$ range of 50–1,000 under dry gas flow at 6.0 l min⁻¹, a dry temperature of 180 °C, a nebulizer pressure of 1 bar, and probe voltage of 14.5 kV.

**Proportion of flavonoids inside and outside of cells**. For the detection of flavonoids in the culture supernatant, the yeast cells were collected by centrifugation at 13,000 × $g$ for 10 min and the supernatant was analyzed by HPLC. For the detection of flavonoids within the cells, the yeast cells were washed twice with double distilled water, disrupted on ice in 30 ml of 50% methanol using a high-pressure homogenizer (JNBIO), and the cell debris was removed by centrifugation at 13,000 × $g$ for 30 min. The supernatant was collected and analyzed by HPLC. The detection of total flavonoids was conducted as mentioned above.

**NMR analysis**. For the isolation of scutellarein, 6 l of fermentation broth was centrifuged and the cells were extracted with methanol three times. The resulting methanol phase was collected and completely evaporated yielding a dry residue. The residue was dissolved in 50 ml methanol and further purified on an Agilent 1260 preparative HPLC with MWD detector (solvent A: 0.1% formic acid; solvent B: methanol; solvent A/B = 45:55; flow rate 10 ml min⁻¹). A reverse-phase C18 column (21.2 × 250 mm, 5 μm, Welch, China) was used for the preparation. ¹H and ¹³C NMR spectra were obtained on a 400 MHz Bruker Avance III spectrometer in DMSO-d6. Chemical shifts ($\delta$) were expressed in p.p.m. and coupling constants ($J$) in hertz (Hz). The results were listed as follows: ¹H NMR (400 MHz, DMSO-d6): $\delta$ 12.81 (s, 1 H, 5-OH), 10.16 (s, 1 H, 7-OH), 7.92 (d, 2 H, $J$ = 8.9 Hz, H-2′, 6′), 6.93 (d, 2 H, $J$ = 8.8 Hz, H-3′, 5′), 6.75 (s, 1 H, H-3), 6.59 (s, 1 H, H-8); ¹³C NMR (100 MHz, DMSO-d6): $\delta$ 182.5 (C-4), 163.9 (C-2), 161.4 (C-4′), 153.8 (C-7), 150.1 (C-8a), 147.5 (C-5), 129.6 (C-6), 128.8 (C-2′, 6′), 121.9 (C-1′), 116.4 (C-3′, 5′), 104.4 (C-4a), 102.7 (C-3), 94.3 (C-8); ESI-MS: $m/z$ 287.0550 [M + H]⁺.

**Fermentation**. The fermentation media used for this work were based on the previous study[58]. The trace metal solution contained: 5.75 g l⁻¹ $ZnSO_4 \cdot 7H_2O$, 0.32 g l⁻¹ $MnCl_2 \cdot 4H_2O$, 0.47 g l⁻¹ $CoCl_2 \cdot 6H_2O$, 0.48 g l⁻¹ $Na_2MoO_4 \cdot 2H_2O$, 2.9 g l⁻¹ $CaCl_2 \cdot 2H_2O$, 2.8 g l⁻¹ $FeSO_4 \cdot 7H_2O$, and 80 ml l⁻¹ 0.5 M EDTA, pH 8.0. The vitamin solution contained: 0.05 g l⁻¹ biotin, 1 g l⁻¹ calcium pantothenate, 1 g l⁻¹ nicotinic acid, 25 g l⁻¹ inositol, 1 g l⁻¹ thiamine HCl, 1 g l⁻¹ pyridoxal HCl, and 0.2 g l⁻¹ *p*-aminobenzoic acid. The fed-batch fermentation medium contained: 20 g l⁻¹ glucose, 15 g l⁻¹ $(NH4)_2SO_4$, 8 g l⁻¹ $KH_2PO_4$, 6.2 g l⁻¹ $MgSO_4 \cdot 7H_2O$, 12 ml l⁻¹ vitamin solution, 10 ml l⁻¹ trace metal solution, and 2.5 g l⁻¹ adenine. The seed medium was same as the fed-batch fermentation medium. All the medium components were purchased from Solarbio. HCl and $H_2SO_4$ were purchased from Sinopharm, China.

Glycerol stocks of strain ΔMC-FU-FC-AAA were used to inoculate 40 ml of SD medium in a 250 ml shake flask and incubated at 30 °C under constant orbital shaking at 220 r.p.m. for 24 h. The entire culture volume was transferred into 400 ml of fresh seed medium and incubated at 30 °C and 220 r.p.m. for 36 h. The seed medium was used to inoculate 2 l of fermentation medium in a Biotech-5JG bioreactor (Baoxing, China) with a maximal working volume of 3 l. The $OD_{600}$ of the culture was monitored and the reactors were inoculated to an initial $OD_{600}$ of 1.2–1.5. The vitamin solution, the trace metal solution, and adenine were added into the autoclaved medium after filter sterilization. The fermentations were performed at 30 °C. The pH was maintained at 5.0 with automatic addition of ammonium hydroxide or 1 M $H_2SO_4$. The agitation rate was kept between 300 and 800 r.p.m., and the air flow was set to 2 l min⁻¹. The DOC was kept above 40%.

The feed was active until residual ethanol produced from the glucose phase was completely depleted. A quasi-exponent feed strategy was adopted as described by Nielsen et al.[59]. The feed contained: 386 g l⁻¹ glucose, 9 g l⁻¹ $KH_2PO_4$, 5.12 g l⁻¹ $MgSO_4 \cdot 7H_2O$, 3.5 g l⁻¹ $K_2SO_4$, 0.28 g l⁻¹ $Na_2SO_4$, 5 g l⁻¹ adenine, 12 ml l⁻¹ vitamin solution, and 10 ml l⁻¹ trace metal solution. The vitamin solution, adenine, and trace metal solution were added into the feed solution in the same way as into the fermentation medium. Samples were taken at regular intervals to measure at $OD_{600}$, and their remaining volume was mixed with an equal volume of absolute methanol, ultrasonicated for 30 min and centrifuged at 13,000 × $g$ for 10 min. The supernatant was stored at −20 °C until HPLC analysis.

**Data availability**. The raw data that supports this study are available from the corresponding authors (Huifeng Jiang and Shengchao Yang) upon reasonable request.

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

## Acknowledgements

We thank Professors Qinhong Wang and Yin Li from Tianjin Institute of Industrial Biotechnology for reading the manuscript and for their helpful comments. We thank Dr. Yibin Zhuang and Yi Cai from Tianjin Institute of Industrial Biotechnology for NMR analysis. This work was supported by grants from the 973 Program (2015CB755704), the National Natural Science Foundation of China (NSFC; Grant Number 31670100), the Key Research Program of the Chinese Academy of Science (Grant No. KFZD-SW-215) and The Hundred Talents Program of the Chinese Academy of Sciences to H.J., as well as grants by the NSFC (81560621), the Major Science and Technique Programs in Yunnan Province (2017ZF002), and the project of young and middle-aged talent of Yunnan province (2014HB011) to S.Y.

## Author contributions

X.L., J. Cheng, and H.J. conceived, designed, and drafted the manuscript. X.L. designed and performed the experiments with the help from G.Z., W.D., L.D., X.C., and J.R. J. Cheng performed the bioinformatics analysis with the help from J.Y., L.K., W.F., J. Chen.,

G.L., and Y.Z. H.J., J. Cai, W.W., Y.M., Y.D., and S.Y. outlined the structure and reviewed the manuscript. All authors read and approved the final manuscript.

## Additional information

**Competing interests:** The authors declare no competing financial interests.

