## [Peer Review File · Nature Communications]

Reviewers' comments:

Reviewer #1 (Remarks to the Author):

The manuscript details research undertaken into the completion of our understanding of the biosynthesis of the flavonoids scutellarin and apigenin-7-O-glucuronide. This was achieved through the identification of the two enzymes responsible for the conversion of apigenin into these compounds after whole genome sequencing of the higher plant *Erigeron breviscapus*. By cloning the genes for these two enzymes together with the genes encoding the pathway for apigenin into yeast, the authors were able to develop a recombinant system for the production of scutellarin and apigenin-7-O-glucuronide. This was then optimized through modification to ensure greater conversion of glucose into malonyl CoA for transformation into flavonoids. Overall the work has been carefully undertaken. The authors have done things in a logical fashion and sound conclusions have been drawn. The pathway for the synthesis was as expected. Apigenin has previously been produced in both yeast and *E. coli* previously – so it is not clear why the authors decided upon yeast for this work.

The genome of *E. breviscapus* has been sequenced (at least in part) previously. Were the authors not able to identify the genes from that previously published work?

The authors state that F7GAT has been reported in many other plants – but do not give any references. How does the enzyme they make recombinantly from their host compare to these others? Are there any kinetic parameters?

Does the F6H gene product not require a reductase? Did the authors not purify and characterize this enzyme as they did with F7GAT?

The yields of the final “breviscapine” extract from yeast appear to be good but this needs to be put into the context of what similar yields have been produced of related materials through other engineering approaches.

Reviewer #2 (Remarks to the Author):

The manuscript by Liu et. al. describes the discovery of the enzymes responsible for the last two steps in the biosynthesis of scutellarin, and apigenin -7-O-glucuronide, the two main ingredients in breviscapine, a Chinese ancient medicinal extract used for the treatment of cardiovascular and cerebrovascular diseases. To achieve this, the authors sequenced the genome of *Erigeron breviscapus*, the plant from which breviscapine is obtained. They then carried out a clever

genomic comparison with four other sequenced plants that can also produce scutellarin, and eight that cannot. The authors used the yeast *Saccharomyces cerevisiae* to test their hypotheses, and ultimately to develop a microbial host to produce scutellarin and apigenin-7-O-glucuronide via fermentation. The study is a valuable demonstration of how comparative genomics is a powerful strategy to elucidate biosynthetic pathways of valuable natural products, and reminds readers of the study by Ro et al (Nature 2006) in which they used comparative genomics and yeast metabolic engineering to produce the antimalarial drug precursor artemisinin acid. The authors reference this study by Ro et al, but miss the opportunity to compare it with their strategy.

This study is important and will be valuable to the natural product and metabolic engineering communities. However, there are some problems that need to be addressed before it can be considered for published in Nature Communications.

1) The manuscript is severely lacking in specifics of plasmid construction and yeast strain development. All plasmids used, either from previous studies or developed in this study, must be well described. This includes providing the vector type (2micron, centromeric, or integrative), the marker used, and the promoters used to express all genes that are overexpressed. A table listing these characteristics of all plasmids in this study should also be included (it is useful to name plasmids for this purpose). Similarly, the yeast strains used should be described in detail, including their genetic background, and any new deletions, integrations, or plasmid transformations carried out in this study. The authors say little besides the fact that they use a W303 strain. There should also be a table describing yeast strains.

2) In lines 118-120, the authors say that 10% of apigenin-7-O-glucuronide diffuses out the cell. This is unlikely to be the case given the sugar moiety in this molecule. Most likely it is actively pumped out of the cell.

3) In a related issue, the authors claim that EbF6H acts first to convert apigenin to scutellarein and then EbF7GAT converts scutellarein to scutellarin. Their claim is based on not observing scutellarin formation when apigenin-7-O-glucuronide is added to the growth media of yeast expressing EbF6H, but observing scutellarein formation when they feed apigenin in the media. However, this observation can also be explained if yeast is unable to import apigenin-7-O-glucuronide, which will not likely diffuse through the membrane due to its sugar moiety, and would likely need a transporter to be imported. Therefore, the strict enzymatic order that the authors claim is not well substantiated by their experimental observations. To be able to make this claim they need a control to show that yeast is able to import apigenin-7-O-glucuronide in enough quantities that EbF6H would have the chance to convert it if it accepted it as a substrate. Alternatively, they could get in vitro evidence that EbF6H is able to convert apigenin but not apigenin-7-O-glucuronide. The possibility remains that both enzymes are promiscuous enough to convert either set of substrates as shown in figure 1A, but yeast is simply unable to import

apigenin-7-O-glucoronide.

4) On line 97 the authors state that "...F7GAT has been reported in many other plants...", but provide no references. Some references must be added to this comment.

5) Although the manuscript is generally well written, they need to do a thorough grammatical revision, including of the supplementary information. For example,

- In line 138 "production may be an autapomorphy restricted to the Astraceae family" is missing a verb.
- Last line in "The history of breviscapine" in supplementary information reads: "Therefore more 10 million patients would use breviscapine injections each year in China", which is very confusing
- The last sentence in the second to last paragraph of supplementary information reads: "The dissolved oxygen concentration was controlled above 40% throughout binding the agitation rate". Again this sentence is poorly constructed and confusing
- This is not an extensive list of errors. The authors should go through the entire manuscript carefully to correct this type of errors.

6) Figure legend 2C is not descriptive enough. It does not explain what the color scale in the heat map represents.

7) The initial yield of breviscapine yeast is characterized as "very low" (line 189), but no quantity is given. How much did this initial strain produce (in mg/L)?

8) They need to provide a more detailed method of how they homogenized plant tissue and separated plant cells for flow cytometry experiments.

9) The first paragraph in the "Gene integration and knockout" section in Supplementary Information seems out of place and uninformative

10) They state in early measurements that 10% of product is found in supernatant and the rest is inside the cell. However in all later reports of titers they don't break down how much is found in supernatant, and how much is in the cells, which they need to do. Also, they don't provide a detailed protocol for measuring products from the cytosol versus products inside the cells. There is just one method lumped together. They need to be more specific.

Reviewer #3 (Remarks to the Author):

This study beautifully combined the power of genomics and biochemistry to illustrate the biosynthetic pathway of breviscapine. I think the paper is publishable in Nature Communications after revision.

Major comments:

1. The genome size of *E. breviscapus* was estimated to be 1.52 Gb by flow cytometry. It can also be estimated based on K-mer analysis using Illumina reads. Here, 1.2 Gb of the assembled genome is obtained. In this manuscript, the genes involved in the biosynthetic pathway of breviscapine were identified on the genome scale based on the genome assembly. Therefore, it is necessary to assess the completeness of the genic regions (BUSCO and RNA-seq) in this assembly.

2. EbF7GAT's function.

How about the kinetic parameter of EbF7GAT in the catalysis of apigenin-7-O-G or scutellarin? The substrate specificity of EbF7GAT on other structurally similar flavonoids?

There are many other reported F7GATs, such as those such as those mentioned in the paper Akio Noguchi, et al. *The Plant Cell*, 2009, 21, 1556-1572; Eiichiro Ono, et al. *Phytochemistry*, 2010, 71, 726-735. Please analyze their evolutionary relationship by integrating more F7GATs.

3. EbF6H's function

In Fig. 3B, it looks like EbF6H is able to oxidize apigenin-7-O-G to generate scutellarin, which is not the case described in the following paragraph. So I suggest add more sentences to explain how scutellarin is produced in the preliminary screen experiment to avoid this confusion.

Similar again for Fig. 3E. CYP706Xs are not apigenin-7-O-G oxidase, right?

4. Yeast engineering

Are the pathway enzymes, e.g. C4H, CHS, PAL, used in the engineering yeast from *E. breviscapus* or other species? If these enzymes were patented by other groups, this may cause potential conflict especially when you emphasize in this study to use engineering yeast as an alternative resource to produce breviscapine. If they are from *E. breviscapus*, the prefix 'Eb' could be added to gene names to avoid this confusion.

Other minor comments:

1, Searching for EbF6H also has been done by other groups. Please mention the paper (Jiang N.H., et al. *PloS One*, 2014, 9, e100357) in the text.

2, Line 97, change 'F7GAT has' to 'F7GATs have'

3, Line 180, change 'Supplementary Fig. 5' to 'Supplementary Fig. 3'

4, Line 185-186, informal presentation. Please rewrite this sentence.

5, Stretched fonts in several figures used in the main text.

6, discussion part?

Authors' responses to reviewers' comments

Reviewers' comments:

Reviewer #1 (Remarks to the Author):

Comments: The manuscript details research undertaken into the completion of our understanding of the biosynthesis of the flavonoids scutellarin and apigenin-7-O-glucuronide. This was achieved through the identification of the two enzymes responsible for the conversion of apigenin into these compounds after whole genome sequencing of the higher plant *Erigeron breviscapus*. By cloning the genes for these two enzymes together with the genes encoding the pathway for apigenin into yeast, the authors were able to develop a recombinant system for the production of scutellarin and apigenin-7-O-glucuronide. This was then optimized through modification to ensure greater conversion of glucose into malonyl CoA for transformation into flavonoids.

Overall the works has been carefully undertaken. The authors have done things in a logical fashion and sound conclusions have been drawn. The pathway for the synthesis was as expected. Apigenin has previously been produced in both yeast and *E. coli* previously – so it is not clear why the authors decided upon yeast for this work.

Responses: We thank the reviewer for his/her comments. Both *E. coli* and yeast (*S. cerevisiae*) are classical platforms for natural compounds production. However the GRAS (generally recognized as safe) status of yeast could facilitate subsequent application for the production of pharmaceuticals and nutraceuticals. The eukaryotic nature of yeast may facilitate functional expression of plant-derived heterologous genes, such as cytochrome P450 enzymes. In this work, we had to express several P450 enzymes such as C4H, FSII and F6H. In future we also want to use the engineered yeast to produce breviscapine as drug at industrial scales. Hence, yeast is a better platform than *E. coli* for our work.

Comments: The genome of *E. breviscapus* has been sequenced (at least in part) previously. Were the authors not able to identify the genes from that previously published work?

Responses: In order to facilitate other researchers to use the genome data without restriction, we released *Erigeron breviscapus* genome data in *Gigascience* used article type Data Note (https://academic.oup.com/gigascience/pages/data_note). One of the aims of a Data Note is to incentivize and more rapidly release data before subsequent detailed analysis has been carried out. This is similar to the genomic data of rock pigeon, which was firstly released in *GigaScience* in July 2011, but the genomic analysis paper of rock pigeon was published on *Science* in February 2013.

Comments: The authors state that F7GAT has been reported in many other plants – but do not give any references. How does the enzyme they make recombinantly from their host compare to these others? Are there any kinetic parameters?

Responses: Thanks. We added references (Akio Noguchi, *et al.* The Plant Cell, 2009, 21, 1556-1572; Eiichiro Ono, *et al.* Phytochemistry, 2010, 71, 726-735.) in this version (line 106, page 4 in revised version). The kinetic parameters of purified *EbF7GAT* were determined for both substrates, revealing K_m values of 9.24 and 70.15 μM and k_{cat} of 0.57 s^{-1} and 0.24 s^{-1} , respectively (Supplementary Table 5 and line 113-115, page 4-5 in revised version)

Comments: Does the F6H gene product not require a reductase? Did the authors not purify and characterize this enzyme as they did with F7GAT?

Responses: Thanks. We have listed the sequence of *EbCPR* in the supplementary table 13, and added the relevant content of *EbCPR* in the main text (line 167,177, page 6 and line 186, 188, page 7 in revised version). By isolating microsomal proteins from the strain which expressed both *EbF6H* and *EbCPR* genes, we confirmed the function of *EbF6H* *in vitro*. The microsomal proteins containing proteins of *EbF6H* and *EbCPR* could catalyze the hydroxylation of apigenin at position six, but could not convert hydroxyl group to apigenin-7-O-glucuronide. (Fig. 3d,e and line 186-189, page 7 and 585-608, page 19 in revised version)

Comments: The yields of the final “breviscapine” extract from yeast appear to be good but this need to be put into the context of what similar yields have been produced of related materials through other engineering approaches.

Responses: We compared the yield of breviscapine with other flavonoids like naringenin and apigenin in discussion part. (line 266-267, page 9 in revised version)

Reviewer #2 (Remarks to the Author):

Comments: The manuscript by Liu et. al. describes the discovery of the enzymes responsible for the last two steps in the biosynthesis of scutellarin, and apigenin-7-O-glucuronide, the two main ingredients in breviscapine, a Chinese ancient medicinal extract used for the treatment of cardiovascular and cerebrovascular diseases. To achieve this, the authors sequenced the genome of *Erigeron breviscapus*, the plant from which breviscapine is obtained. They then carried out a clever genomic comparison with four other sequenced plants that can also produce scutellarin, and eight that cannot. The authors used the yeast *Saccharomyces cerevisiae* to test their hypotheses, and ultimately to develop a microbial host to produce scutellarin and apigenin-7-O-glucuronide via fermentation. The study is a valuable demonstration of how comparative genomics is a powerful strategy to elucidate biosynthetic pathways of valuable natural products, and reminds readers of the study by Ro et al (Nature 2006) in which they used comparative genomics and yeast metabolic engineering to produce the antimalarial drug precursor artemisinic acid. The authors reference this study by Ro et al, but miss the opportunity to compare it with their strategy.

This study is important and will be valuable to the natural product and metabolic engineering communities. However, there are some problems that need to be

addressed before it can be considered for published in Nature Communications.

Responses: Thanks for your positive comments. Comparing with previous studies, we made two novelties. One is that we used the genome sequence of *Erigeron breviscapus* to identify all genes in the flavonoid biosynthesis pathways from phenylalanine to scutellarin. The other is that we synthesized two major compounds in breviscapine in one single engineered yeast. It provides us an opportunity to directly produce the drug of breviscapine in yeast in future.

Comments: 1) The manuscript is severely lacking in specifics of plasmid construction and yeast strain development. All plasmids used, either from previous studies or developed in this study, must be well described. This includes providing the vector type (2micron, centromeric, or integrative), the marker used, and the promoters used to express all genes that are overexpressed. A table listing these characteristics of all plasmids in this study should also be included (it is useful to name plasmids for this purpose). Similarly, the yeast strains used should be described in detail, including their genetic background, and any new deletions, integrations, or plasmids transformations carried out in this study. The authors say little besides the fact that they use a W303 strain. There should also be a table describing yeast strains.

Responses: Thank you very much. Now the details of plasmid construction and yeast strain development have been described in methods. (line 464-546, page 15-17 in revised version) We have listed all the detailed content as the reviewer pointed out in the supplementary table 7.

Comments: 2) In lines 118-120, the authors say that 10% of apigenin-7-O-glucuronide diffuses out the cell. This is unlikely to be the case given the sugar moiety in this molecule. Most likely it is actively pumped out of the cell.

Responses: Thank you very much for indicating this mistake. We have revised the sentence to “After five days of fermentation, the engineered yeast produced about 20 mg/L of apigenin-7-O-glucuronide, of which 26.60% was located within the cells and 73.40% had diffused into the culture broth (Supplementary Table 8).” (line 133-135, page 5 in revised version)

Comments: 3) In a related issue, the authors claim that EbF6H acts first to convert apigenin to scutellarein and then EbF7GAT converts scutellarein to scutellarin. Their claim is based on not observing scutellarin formation when apigenin-7-O-glucuronide is added to the growth media of yeast expressing EbF6H, but observing scutellarein formation when they feed apigenin in the media. However, this observation can also be explained if yeast is unable to import apigenin-7-O-glucuronide, which will not likely diffuse through the membrane due to its sugar moiety, and would likely need a transporter to be imported. Therefore, the strict enzymatic order that the authors claim is not well substantiated by their experimental observations. To be able to make this claim they need a control to show that yeast is able to import apigenin-7-O-glucuronide in enough quantities that EbF6H would have the chance to convert it if it accepted it as a substrate. Alternatively, they could get in vitro evidence

that EbF6H is able to convert apigenin but not apigenin-7-O-glucuronide. The possibility remains that both enzymes are promiscuous enough to convert either set of substrates as shown in figure 1A, but yeast is simply unable to import apigenin-7-O-glucuronide.

Responses: Thanks a lot for your suggestion. As you pointed out, we confirmed the function of F6H *in vitro* by extracting the microsomal proteins which contain proteins *EbF6H* and *EbCPR*. Our results show that the *EbF6H* is able to use apigenin as substrate but not apigenin-7-O-glucuronide. (Fig. 3d,e and line 186-189, page 7 and 585-608, page 19 in revised version)

Comments: 4) On line 97 the authors state that “...F7GAT has been reported in many other plants...”, but provide no references. Some references must be added to this comment.

Responses: We have added references in the revised version. (line 105, page 4 in revised version)

Comments: 5) Although the manuscript is generally well written, they need to do a thorough grammatical revision, including of the supplementary information. For example,

- In line 138 “production may an autapomorphy restricted to the Astraceae family” is missing a verb.

Responses: We have revised the sentence to “These pieces of evidence indicated that the encoding gene of F6H in *E. breviscapus* may be an Asteraceae-specific gene.” (line 148-150, page 6 in revised version).

- Last line in “The history of breviscapine” in supplementary information reads: “Therefore more 10 million patients would use breviscapine injections each year in China”, which is very confusing

Responses: We have revised as “In 2016, approximately 72 million breviscapine injections were produced in China. Considering one patient need seven breviscapine injections for one period treatment, we inferred that the products of breviscapine injections benefit more than 10 million patients each year in China.” (Supplementary text in revised version)

- The last sentence in the second to last paragraph of supplementary information reads: “The dissolved oxygen concentration was controlled above 40% throughout binding the agitation rate”. Again this sentence is poorly constructed and confusing

Responses: “The dissolved oxygen concentration was kept above 40%.” (line 677, page 21 in revised version)

- This is not an extensive list of errors. The authors should go through the entire manuscript carefully to correct this type of errors.

Responses: We thank a lot for your excellent and constructive comments. We have tried our best to comply with all of the suggestions from you. Now, the

manuscript has been substantially revised. We also asked native speaker to purify this version. Thanks a lot!

Comments: 6) Figure legend 2C is not descriptive enough. It does not explain what the color scale in the heat map represents.

Responses: We have explained the color scale in the heat map represents in figure legend 2c. (line 910-913, page 30 in revised version)

Comments: 7) The initial yield of breviscapine yeast is characterized as “very low” (line 189), but no quantity is given. How much did this initial strain produce (in mg/L)?

Responses: “Although we were able to produce both components of breviscapine when we introduced all genes of the corresponding biosynthetic pathways into yeast, the yield was still very low, with only 9.2 mg of scutellarin per gram of dry cell weight.” We listed this yield in line 195-197, page 7.

Comments: 8) They need to provide a more detailed method of how they homogenized plant tissue and separated plant cells for flow cytometry experiments.

Responses: We have provided a detailed method of how they homogenized plant tissue in “Methods” part of “Plant materials and genome size estimation”. “The genome size of *E. breviscapus* was estimated by flow cytometry as follows: (1) a small amount of leaves (typically 200 mg) from *E. breviscapus* was placed at the center of a petri dish, washed with deionized water three times and dried with filter paper; (2) 1 ml buffer I (0.1 M citric acid (Sigma-Aldrich), 0.5% Tween 20 (V/V) (Sigma-Aldrich)) was added to the dish and it was made sure that the leaves are submerged, after which the leaves were chopped in the buffer using a fresh razor blade; (3) the supernatant was carefully transferred into a 1.5-mL centrifuge tube and the cell suspension was marked as sample 1; (4) a cell suspension of *Oryza sativa* Nipponbare was used as internal standard and marked as sample 2; (5) 0.5 ml of sample 1 were pipetted and mixed with 0.5 ml of sample 2 to homogeneity, and marked as sample 3; (6) 0.5 ml of solution II (0.4 M Na₂HPO₄·12H₂O (Solarbio, China)) was added to sample 1 and sample 2, 1 ml of solution II was added to sample 3, and all were mixed gently; (7) A final concentration of 50 mg/ml propidium iodide (PI; Solarbio) and 50 mg/ml of RNase (Generay, China) were added to each sample, and shaken gently; (8) The samples were incubated for 20 minutes in the dark with occasional gentle shaking; (9) The samples were carefully put into the flow cytometer to analyze the nuclear DNA content. The results showed that the genome size of *E. breviscapus* was approximately 1.52 Gb.” (line 290-308, page 10 in revised version)

Comments: 9) The first paragraph in the “Gene integration and knockout” section in Supplementary Information seems out of place and uninformative.

Responses: Thanks. We have putted it in the “Chemicals, media, and strain

cultivation” section. (line 458-462, page 15 in revised version)

Comments: 10) They state in early measurements that 10% of product is found in supernatant and the rest is inside the cell. However in all later reports of titers they don't break down how much is found in supernatant, and how much is in the cells, which they need to do. Also, they don't provide a detailed protocol for measuring products from the cytosol versus products inside the cells. There is just one method I μ Mped together. They need to be more specific.

Responses: **We are very sorry to make this mistake. There is 9.1% of apigenin diffuses out the cell and 73.40% of apigenin-7-O-glucoronide diffuses out the cell as listed in Supplementary Table 8. We apologize about this deeply and we have checked all of our data carefully. A detailed protocol was showed in “Methods” part of “Proportion of flavonoids inside and outside of cells”. (line 634-641, page 20 in revised version)**

Reviewer #3 (Remarks to the Author):

This study beautifully combined the power of genomics and biochemistry to illustrate the biosynthetic pathway of breviscapine. I think the paper is publishable in Nature Communications after revision.

Major comments:

Comments: 1. The genome size of *E. breviscapus* was estimated to be 1.52 Gb by flow cytometry. It can also be estimated based on K-mer analysis using IlluMina reads. Here, 1.2 Gb of the assembled genome is obtained. In this manuscript, the genes involved in the biosynthetic pathway of breviscapine were identified on the genome scale based on the genome assembly. Therefore, it is necessary to assess the completeness of the genic regions (BUSCO and RNA-seq) in this assembly.

Responses: **Thank you for your suggestion, we have added the assessment in the revised manuscript (line 85-88, page 4 in revised version). The total genome assembly amounted to 1.2 Gb, which is ~80% of the estimated total genome size. CEGMA was used to evaluate the quality of the final assembly with a set of 248 ultra-conserved core eukaryotic genes. Comparison analysis showed that 217 of 248 genes (87.5 %) could be fully annotated in our genome assembly. Although the coverage of genome is not very high, we identified all genes in the biosynthetic pathway of scutellarin.**

Comments: 2. EbF7GAT's function.

How about the kinetic parameter of EbF7GAT in the catalysis of apigenin-7-O-G or scutellarin?

The substrate specificity of EbF7GAT on other structurally similar flavonoids?

There are many other reported F7GATs, such as those such as those mentioned in the paper Akio Noguchi, et al. The Plant Cell, 2009, 21, 1556-1572; Eiichiro Ono, et al. Phytochemistry, 2010, 71, 726-735. Please analyze their evolutionary relationship by integrating more F7GATs.

Responses: The kinetic parameters of purified EbF7GAT for both substrates were determined, where K_m are 9.24 μM and 70.15 μM and k_{cat} are 0.57s⁻¹ and 0.24s⁻¹ respectively. (line 113-115, page 4-5 in revised version) We also found that EbF7GAT can catalyze many other flavonoids such as narigenin, kaempferol, quercetin, chrysin, baicalein, luteolin, diosmetin, chrysoeriol effectively. (line 116-118, page 5 in revised version) At the same time, we analyzed the evolutionary relationship of F7GATs within and between species. Totally 83 glycosyltransferase genes from *E. breviscapus* genome were identified and assigned into 15 gene families (Supplementary Fig. 1). Based on phylogenetic tree of gene family UGT88 which typically transfer UDP-glucuronic acid to C7 position of flavonoids, we identified *EbF7GAT* in *E. breviscapus* (Supplementary Fig. 2a).

Comments: 3. EbF6H's function

In Fig. 3B, it looks like EbF6H is able to oxidize apigenin-7-O-G to generate scutellarin, which is not the case described in the following paragraph. So I suggest add more sentences to explain how scutellarin is produced in the preliminary screen experiment to avoid this confusion.

Similar agin for Fig. 3E. CYP706Xs are not apigenin-7-O-G oxidase, right?

Responses: We have added more experimental results about the microsomal enzyme reaction experiment, and confirmed that *EbF6H* is unable to oxidize apigenin-7-O-G to generate scutellarin (Fig. 3). In Fig. 3B, the starting strain A1 could produce excess apigenin, a part of apigenin could not be converted into scutellarein and catalyzed by EbF7GAT to form scutellarin. Another part of apigenin could not be converted into scutellarein immediately and catalyzed by EbF7GAT to form apigenin-7-O-glucoronide. We confirmed the function of F6H in vitro by extracting the microsomal proteins which contain proteins EbF6H and EbCPR. Our results show that the EbF6H is able to use apigenin as substrate but not apigenin-7-O-glucoronide. (Fig. 3d,e and line 186-189, page 7 and 585-608, page 19 in revised version)

Comments: 4. Yeast engineering

Are the pathway enzymes, e.g. C4H, CHS, PAL, used in the engineering yeast from *E. breviscapus* or other species? If these enzymes were patented by other groups, this may cause potential conflict especially when you emphasize in this study to use engineering yeast as an alternative resource to produce breviscapine. If they are from *E. breviscapus*, the prefix 'Eb' could be added to gene names to avoid this confusion.

Responses: Thanks for your suggestions. The enzymes in the pathway are all from *E. breviscapus*. The prefix 'Eb' has been added to gene names.

Other minor comments:

Comments: 1, Searching for EbF6H also has been done by other groups. Please mention the paper (Jiang N.H., et al. PloS One, 2014, 9, e100357) in the text.

Responses: Thanks. We have referenced this paper in the text. (line 70, page 3)

Comments: 2, Line 97, change 'F7GAT has' to 'F7GATs have'

Responses: The sentence has been changed into "Flavonoid-7-O-glucuronosyltransferases (F7GATs), which typically transfer UDP-glucuronic acid to C7 position of flavonoids, have been reported in *Lamiales*, but the functional gene had not yet been confirmed in *E. breviscapus*." (line 103-105, page 4)

Comments: 3, Line 180, change 'Supplementary Fig. 5' to 'Supplementary Fig. 3'

Responses: The sentence has been changed into "since the EbF6H orthologs in the studied Asteraceae species (CcF6H, LsF6H, CcarF6H, CtF6H) also showed the same function (Supplementary Fig. 8)." (line 247-249, page 8)

Comments: 4, Line 185-186, informal presentation. Please rewrite this sentence.

Responses: Thanks. We changed into "our study illustrates the power of genomic data, since it would be extremely difficult to pinpoint a single novel gene out of several hundred P450 candidates without the genome sequence." and removed this sentence into discussion part. (line 253-255, page 9)

Comments: 5, Stretched fonts in several figures used in the main text.

Responses: Thank you very much. All the fonts of figures in the main text have been revised.

Comments: 6, discussion part?

Responses: Thanks. We changed the format of MS and added the discussion part in this version. (line 232-281, page 8-9)

Reviewers' Comments:

Reviewer #1 (Remarks to the Author):

All the major comments appear to have been addressed. This is a paper that will be of interest to a brand range of people working on synthetic biology and metabolic engineering.

Reviewer #2 (Remarks to the Author):

The authors addressed most issues brought up by Reviewer 2, with one exception regarding comment 2 of Reviewer 2: They still use the term “diffused”, which has a very specific physical meaning, that is unlikely to occur across the cell membrane, given the sugar moiety of the product. They should use the more general term “secreted”. Other than that, the manuscript is ready for publication

Reviewer #3 (Remarks to the Author):

All my recommendations and comments have been acted properly upon. I have no further requirements for the authors.